# Quantitively Characterizing the Chemical Composition of Tailored Bagasse Fiber and Its Effect on the Thermal and Mechanical Properties of Polylactic Acid-Based Composites

**DOI:** 10.3390/polym11101567

**Published:** 2019-09-26

**Authors:** Haoqun Hong, Ruijing Xiao, Quannan Guo, Hao Liu, Haiyan Zhang

**Affiliations:** 1School of Materials and Energy, Guangdong University of Technology, Guangzhou 510006, China; 2Guangdong Provincial Key Laboratory of Functional Soft Condensed Matter, Guangdong University of Technology, Guangzhou 510006, China

**Keywords:** biocomposite, natural fibers, interface, microstructural analysis

## Abstract

Natural fiber reinforced polymer-based composites have been growing into a type of green composites. The properties of natural fiber reinforced polymer-based composites are closely related to the structure of natural fibers. Bagasse fiber (BF) is one of the most used natural fibers for preparing natural fiber reinforced polymer-based composites. However, few examples of previous research touch on the quantitatively characterization of structure of BF and its effect on the properties of BF reinforced polymer-based composites. In this work, four kinds of BF including untreated BF (UBF), alkali treated BF (ABF), BF modified by silane coupling agent (SBF), and BF modified combining alkali treatment with silane coupling agent (ASBF) were prepared and melting blended with polylactic acid (PLA) to prepare PLA/BF composites. Fourier transform infrared spectroscopy (FTIR), X-ray diffraction (XRD), scanning electron microscopy (SEM), X-ray photoelectron spectroscopy (XPS), thermogravimetry (TGA) and mechanical properties testing were used to characterize and analyze the structure and properties of modified BF and its reinforced PLA-based composites. Results showed that the used methods changed the structure of BF and their bonding modes. The surface energies of UBF, ABF, SBF, and ASBF were 19.8 mJ/m^2^, 34.7 mJ/m^2^, 12.3 mJ/m^2^, and 21.6 mJ/m^2^, respectively. The O/C ratios of UBF, ABF, SBF and, ASBF are 0.48, 0.53, 0.47, and 0.51. Due to the synergistic effect of alkali treatment and silane coupling agent modification on the surface chemical properties, the content of silicon elements on the surface of ASBF (4.15%) was higher than that of ASBF (2.38%). However, due to the destroying of alkali treatment on the microstructure of BF, the alkali treatment had no prominently synergetic effect with coupling agent modification on the mechanical properties of PLA/BF composites. Alkali treatment removed the small molecular compounds from BF, decreased its thermal stability, and increased the crystalline region and crystallinity of cellulose. Meanwhile, alkali treatment made BF fibrillated and increased its contactable active area with the coupling agents, but destructed the nature structure of BF. The silane coupling agent played a more important role than alkali treatment did in improving the interfacial compatibility of PLA/BF composites.

## 1. Introduction

Recently, green composites which are made of recycled or degradable polymers and renewable plant resources, have aroused the great interest of researchers due to severe environmental concerns such as increasing energy consumption, greenhouse gas exhaustion, depletion of fossil fuels, and accumulation of plastic waste [1,2].

Polylactic acid (PLA) is one of the most used raw materials for the preparation of green composites [3,4]. It is a thermoplastic aliphatic polyester which has mechanical, thermal, and barrier properties comparable with commercial common plastics [5,6,7]. It is a fully biodegradable biopolymer that can be processed by conventional plastics processing equipment [8]. PLA has found a wide range of applications in packaging [9,10,11], biological medicine [12,13], the automotive industry [14,15,16], textiles [17], disposable utensils [18], construction and building [19,20], and 3D printing [21,22].

Another kind of raw material for the preparation of green composites is natural fiber (NF), especially recycled natural fiber. The recycling of discarded NFs such as wood flour (WF), rice chaff (RF), bagasse fiber (BF), and sisal fiber (SF) has caught more and more people’s attention with the upgrading of environment protection as a priority. Compared with inorganic fibers such as glass fiber and carbon fiber, NF has advantages in terms of light weight, low cost, high strength, high modulus, and biodegradation, and have been widely used to prepare green composites [23,24,25,26].

BF is a kind of NF that is common in tropical and subtropical regions. It is the residue of squeezed sugarcane bagasse. Currently most BF is treated as fertilizer, feed, or fuel. These treatments are not benign to the environment and do not take full advantage of BF. Compounding BF with polymeric materials to prepare natural fiber reinforced polymer-based composites not only realizes the efficient recycling of BF but also improves the integrated properties of polymeric materials and expands their application ranges.

The main components of BF are cellulose (40%–50%), hemicellulose (25%–35%) and lignin (15%–35%) which have many hydroxyls [27]. These hydroxyls have strong polarity and hygroscopicity, which makes BF weak compatible to most polymeric materials at their interfaces when they are fabricated into composites, resulting in the decreasing of mechanical properties of the composites. Meanwhile, these hydroxyls tend to form the intramolecular hydrogen bonds inside the cellulose, which makes BF easily aggregate when BF is compounded with polymers. The inferior dispersion of BF impedes the reinforcement of BF to polymeric materials and limits the applications of BF reinforced polymer-based composites.

Therefore, how to improve the interfacial compatibility between BF and polymeric materials plays a key role in improving the mechanical properties of BF reinforced polymer-based composites [28,29,30,31,32,33,34]. At present, the methods to improve the interfacial compatibility of natural fiber (NF) reinforced polymer-based composites are filling compatibilizers into the composites [32,34] or surface treatment and modification of NF [28,29,30,31,33].

The filled compatibilizers are the strongly interfacial interactive polymers such as the graft copolymers or ammonium lignosulphonate (AL). Csikós [32] used maleic anhydride grafted polylactic acid as the interfacial compatibilizers for PLA/WF composites. Results showed that maleic anhydride grafted polylactic acid significantly enhanced the interfacial compatibility of PLA/WF and evidently increased the mechanical properties of PLA/WF composites. Since maleic anhydride grafted polylactic acid can offer strong and powerful interfacial interactions through chemical bonds, the observed interfacial morphology by scanning electron microscope shows that WF was broken during the stretching of the composites. Hu [34] used AL as interfacial modifiers and investigated the effect AL on the mechanical and thermal properties of PLA/WF composites. Results showed that AL increased the flexural strength, flexural modulus, and internal bond strength of PLA/WF composites when the mass ratio of PLA to BF is 7:3.

The surface treatment and modification of natural fiber use alkali solution to treat natural fiber and to clear up the small molecular compounds that covered on the surface of natural fibers, or use silane coupling agents to modify the surface of natural fiber to organize and functionalize the surface of natural fiber. As a result, the interfacial compatibility between natural fiber and polymeric materials is enhanced. Orue et al. [33] used methods such as the treatment of NaOH and modification of silane coupling agents to modify SF and investigated the effect of various methods on the mechanical properties of PLA/SF composites. Results shows that alkali treatment made the surface of SF became rough and thin, and silane coupling agents made the surface of SF organized, as well as increasing the interfacial compatibility between SF and PLA. These two methods both improved the interfacial compatibility of PLA/SF composites and therefore increased the mechanical properties of the composites at various grades.

Due to the complexity of NF microstructure, how the NF microstructure affects the performances of its reinforced composites is associated with the types of NF used or the microstructure tailoring conditions such as the concentration and time used for alkali treatment and silane modification of NF. At present there is little research on the quantitative characterization of tailored BF microstructure after alkali treatment and silane modification, and its effect on the thermal and mechanical properties of PLA-based composites though other NF reinforced PLA-based composites have been widely reported. In this paper, the methods in terms of alkali treatment, silane coupling agent modification and combination of alkali treatment and silane coupling agent modification were used to modify BF. Fourier transform infrared spectroscopy (FTIR), X-ray diffraction (XRD), scanning electron microscopy (SEM), Polarizing optical microscope (POM), thermogravimetry (TGA), differential scanning calorimetry (DSC), and mechanical properties testing were used to investigate the microstructure of modified BF and its effects on the compatibility and properties of PLA/BF composites.

## 2. Materials and Methods

PLA (Nature Works^®^ 4032D in pellet) was purchased from Nature Works LLC. The melting flow rate of PLA is 7.72 g/10 min at 230 °C and 2.16 kg. Bagasse fiber (BF), with an average size of 180 μm and aspect ratio of 2.89, was supplied by Guangzhou sugarcane and the Sugar Research Institute. The major components of BF are 46.1% cellulose, 24.7% hemicellulose, 22.6% lignin, and 6.6% ash and other impurities, as offered by the supplier. Sodium hydroxide (analytical reagent) was purchased from Tianjin Shigeru Chemical Reagent Factory (Tianjin, China). 3-aminopropyltriethoxysilane (APTES) was the analytical reagent that was purchased from Shanghai Aladdin biochemical Polytron Technologies Inc. Acetone (analytical reagent) was purchased from Tianjin Shigeru Chemical Reagent Factory.

BF was dried for 6 h in an 80 °C drying oven and smashed for 5 min in a high-speed disintegrator. The smashed BF was soaped in 3 wt % NaOH/water solution with 1: 20 solid-to-liquid ratios for 4 h under mechanical stirring. The alkali treated BF was rinsed with water 5 times to sufficiently clean NaOH and then filtered. The cleaned BF was air dried for 12 h and then dried for 24 h in a drying oven at 80 °C. The as-prepared BF was designated as alkali treated BF (ABF).

BF was dried for 6 h in a drying oven at 80 °C and smashed for 5 min in a high-speed disintegrator. The smashed BF was soaped in 6 wt % APTES/acetone solution with 1: 20 solid-to-liquid ratios for 48 h. The silane treated BF was eluted with acetone for 5 times to sufficiently clean uncoupled APTES and filtered. The cleaned BF was air dried for 12 h and then dried for 24 h in a drying oven at 80 °C. The as-prepared BF was designated as SBF. The BF prepared successively by alkali treatment and silane treatment described above was designated as alkali treatment with silane coupling agent (ASBF).

Before processing, the untreated bagasse fiber (UBF), ABF, BF modified by silane coupling agent (SBF), BF modified combining alkali treatment with silane coupling agent (ASBF), and PLA were dried in a drying oven at 80 °C. The dried PLA and various BF (UBF, ABF, SBF and ASBF) were uniformly mixed according to the compositions listed in Table 1. The uniformly mixed raw materials were extruded and pelleted in a two-screw extruder at 185 °C to obtain 4 different PLA/BF pellets. The PLA/BF pellets were dried in a drying oven at 80 °C. The dried PLA/BF pellets were injected at 190 °C in an injection molding machine into PLA/BF composites with standard specifications for a tensile test, bending test, and notched impact test. No preferred orientation of the bagasse fibers was set in all PLA/BF composites. The as-prepared PLA/BF composites were designated as PLA/UBF composites, PLA/ABF composites, PLA/SBF composites, and PLA/ASBF composites, respectively. A pristine PLA sample was also prepared under the same conditions for comparison.

The surface contact angle was performed on a Data physics OCA15 surface contact angle analyzer. The samples were compressed into a surface-smooth tablet with diameter of 20 mm and thickness of 2 mm. Distilled water and glycerol were used to measure the surface contact angle. For each sample, 10 different locations were used to measure the surface contact angle. The surface energy was calculated based on the measured surface contact angle according to the Owens–Wendt formula [35,36] as shown in Equation (1):(1)$$\left(1-\cos\theta )\gamma_{L}=2[(\gamma_{L}^{d}\gamma_{S}^{d}{)}^{0.5}+(\gamma_{L}^{p}\gamma_{S}^{p}{)}^{0.5}\right]$$
where *θ* is the contact angle of the fibers with the test liquid, $\gamma_{L}^{d}$, $\gamma_{L}^{p}$ and $\gamma_{L}$ are dispersive, polar and total surface energy of the test liquid, respectively, and $\gamma_{S}^{d}$, $\gamma_{S}^{p}$ are dispersive and polar components of the surface energy of fibers, respectively. The total surface energy $\gamma_{S}$ was calculated by $\gamma_{L}^{d}\gamma_{S}^{p}$.

Chemical structures of the fibers were characterized on an FTIR spectrophotometer (Thermoscientific NICOLET 6700, Thermo Scientific, Waltham, MA, USA). The samples were prepared by mixing the materials and KBr in a proportion of 1:100 (*w*/*w*). For each detection, 16 scans were accumulated with a 4 cm^−1^ resolution.

SEM micrographs were obtained using a FEI Quanta 200 scanning electron microscope (Thermo Scientificcompany, Hillsboro, OR, USA) performing from 15 to 20 kV, using secondary electrons to obtain information of fibers morphology. The PLA/BF samples were brittle broken in liquid nitrogen, fixed to a holder with double-side adhesive carbon tape, and submitted to metallic recovering with gold.

XPS was carried out on an X-ray scanning micro-prober photoelectronic energy spectrometer (250Xi, Thermofisher Scientific Co., Ltd.) by using a monochromatic Al Kα X-ray source (1486.68 eV). The scanning spectra and high-resolution spectra were recorded in the constant pass energy of 150 eV and 50 eV, respectively. High-resolution spectra of C1s, O1s, Si1s, and N1s were measured to quantitatively characterize the binding energy and atomic concentration at a pass energy of 10 eV and resolution of 0.05 eV/step acquired by five scans. The peak differentiation was carried out by using the XPS peak software. The curve differentiating deviation was controlled as little as possible.

The XRD patterns were obtained in a Shimadzu XRD 6000 diffractometer. The measuring conditions were: CuKa radiation with graphite monochromator, 40 kV voltage, and 40 mA electric current. The patterns were obtained in 5–90°2θ angular intervals with 0.05° step and 1 s of counting time.

TGA was carried out on a TA SDT 2960 simultaneous DSC-TGA instrument (TA Instruments, New Castle, DE, USA). 5–10 mg samples were taken out from various BF and their composites, respectively, for analysis. The samples were heated from room temperature to 600 °C at calefactive rates of 10 °C/min under nitrogen condition. The weight loss of the samples was recorded as a function of temperature.

DSC was carried out on a TA DSC 2910 instrument (TA Instruments, New Castle, DE, USA) by heating the samples to 200 °C at calefactive rates of 10 °C/min, holding for 10 min to avoid the thermal history, cooling to 40 °C (with 5 min equilibration at 40 °C) and reheating to 200 °C at constant rates of 10 °C/min under nitrogen condition.

POM micrographs were obtained through a polarizing microscope (BA310Pol, Motic China Group Co., Ltd., Shenzhen, China). A small amount of PLA based composites were put on the slide glass and then transferred to the temperature control platform. The samples were heated to 220 °C, which was maintained until the samples were completely melted. The melted samples were covered by the cover glass and held for 3 min. Then the melted samples were cooled to room temperature at 60 s/°C. The growing crystalline morphology of the cooling samples was captured at various times.

The tensile and flexural tests were performed on a universal testing machine (DCS-5000, Shimadzu, Japan) at room temperature according to ISO527-2: 1993 (with dumbbell sample dimensions of 170 × 10 × 4 mm) and ISO178: 2001 (with sample dimensions of 80×10×4 mm), respectively. A non-notched impact test was performed according to ISO179, with the sample dimensions of 80 × 10 × 4 mm. For each testing, five specimens of composites were analyzed.

## 3. Results

### 3.1. Polarity of the BF Surface

Figure 1 is the comparison of polarity of the BF surface demonstrated by the contact angle and surface energy of the BF surface. The contact angles of UBF, ABF, SBF, and ASBF are 105.2°, 80.7°, 118.4° and 100.9°, respectively. The surface energies of UBF, ABF, SBF, and ASBF are 19.8 mJ/m^2^, 34.7 mJ/m^2^, 12.3 mJ/m^2^, and 21.6 mJ/m^2^. The variation of the contact angle goes against that of the surface energy. A higher contact angle suggests a lower polarity and a lower surface energy at the BF surface [36,37]. The alkali treatment decreases the contact angle of ABF and increases its surface energy, as compared to UBF. The reason for this is that the alkali treatment removed the impurities covering on the surface of BF and thus enhanced the polarity of BF. The contact angle of SBF is greater than that of UBF, suggesting that the silane coupling agents decreased the surface polarity of BF. Note that the surface energy of ASBF is higher than that of ABF. This is attributed to the fact that the alkali treatment coarsened the surface of ASBF, which increased the contact angle of ASBF and thus the surface energy calculated from the contact angle (see the following SEM morphology).

### 3.2. FTIR Characterization of BF

Figure 2 is the FTIR spectra of BF modified by various methods. The spectra show several common peaks that are attributed to BF [38,39]. The peak at 3100 cm^−1^ is attributed to the stretching vibration of C=C in lignin. The peak at 2850cm^−1^ is attributed to the stretching vibration of methylene. The peak at 1700–1750 cm^−1^ is attributed to the stretching vibration of C=O. The peak at 1400–1550 cm^−1^ is attributed to the skeletal vibration of the phenyl group. The peak at 1000–1200 cm^−1^ is attributed to the stretching vibration of C–O. The peaks at 800 cm^−1^ and 850 cm^−1^ are attributed to the out-of-plane bending vibration of C=C in lignin. The peak at 720 cm^−1^ is attributed to the in-plane bending vibration of methylene.

UBF contains a great deal of hemicelluloses oligomers in different chemical environments. The carbonyl groups of hemicelluloses show an obvious peak at 1733 cm^−1^. As for ABF, the peak at 1733 cm^−1^ in ABF is weaker than that in UBF, suggesting that the hemicellulose oligomers and amorphous celluloses in ABF was destroyed or removed by alkali treatment [33,37,40]. Meanwhile, the peak at 1250 cm^−1^ is attributed to aryl-alkyl ethers of lignin, indicating the reduction of lignin in ABF. As for SBF, its infrared spectroscopy (IR) spectrum is similar to that of UBF with an identifiable stretching vibration peak of Si-O-Si and Si–O–C bonds at 700–1300 cm^−1^ [41,42,43,44]. The Si–O–Si bonds come from the cross-linked coupling agents and the Si–O–C bonds come from the chemical reaction between the coupling agents and bagasse [41,42,43]. The peaks that come from Si–O–Si and Si–O–C bonds are overlapped by the peaks of fingerprint region of cellulose. Therefore, it is not easy to recognize the peaks that come from Si–O–Si and Si–O–C bonds. Only the peak at 1204 cm^−1^ that is attributed to Si–O–C bonds is obviously recognized. As for ASBF, the variation of peaks is similar to that of ABF after alkali treatment. But the alkali treatment enhanced the formation of more Si–O–Si and Si–O–C bonds, making the peak at 716 cm^−1^ that is attributed to Si–O–Si stronger.

### 3.3. SEM Characterization of BF

Figure 3 is the SEM morphology of BF modified by various methods. The surface of pristine UBF is tight and absorbed by impurities. The surface of ABF is sculptured by alkali to be fibrillated and deeply grooved, resulting in the increasing of surface area of bagasse. It is evident that the fibrillation of ABF by alkali treatment destroyed the condensed structure of bagasse where some fragment fibers remained [45]. The surface of SBF is covered by the modified silanes and looks smooth. The surface of ASBF do not shows deep grooves but rather coarser grooves than that of SBF due to the removing of impurities and the covering of silane coupling agents on the surface of ASBF.

Table 2 lists the elements distribution on various BF surfaces analyzed based on the observed SEM morphology. The carbon element content of ABF is lower than that of UBF, while the carbon element contents of SBF and ASBF are higher than that of UBF. The oxygen element content of ABF is higher than that of UBF while the oxygen element contents of SBF and ASBF are lower than that of UBF, suggesting that the polarity of ABF is stronger than that of UBF and the polarity of SBF and ASBF is lower than that of UBF. This is because the removing of impurities on the surface of ABF by alkali treatment covering made the ABF more hydrophilic than UBF and the surface organization by the coupling agents made SBF and ASBF more hydrophobic than UBF. Note that the oxygen element content of ASBF is lower than that of SBF, suggesting that the removing of covered impurities made the surface modification on ASBF more impactful than on SBF, as also evidenced by the silicon element content. The silicon element content of ASBF is higher than that of SBF.

### 3.4. XPScharacterization of BF

BF is primarily composed of three elements such as carbon, hydrogen, and oxygen. The major components of BF are similar to those of wood which mainly form cellulose, hemicellulose, lignin and other minor components (tannin and pectin). BF contains silicon after the silane coupling agent treatment. XPS is an effective measure to explore the surface chemicals and elements valence of various treated BF. The results are shown in Figure 4, as corresponding to UBF, ABF, SBF, and ASBF, respectively.

Because of the component similarity, the XPS spectra of BF is similar to that of wood [46,47,48]. The carbon atom on BF surface has five combination modes in terms of C1, C2, C3, C4, and C5. C1, whose electron bonding energy is 285 eV, is attributed to the C-C and C-H single bond which exists in the backbone of cellulose and lignin. C2, whose electron bonding energy is 286.5 eV, is attributed to the C–O single bond which exists in the connection of carbon with hydroxyl of cellulose and lignin and in the ether bond of lignin. C3, whose electron bonding energy is about 288–288.5eV, is attributed to the carbonyl group (C=O) and carbon atom that links with two oxygen atoms (O–C–O) which exists in the backbone of cellulose or in the ketone and aldehyde groups of lignin. Since the alkali treatment removed part of lignin, the C3 peak area of ABF is lower than that of the other BF, as indicated by Table 3. C4, whose electron bonding energy is about 289-289.5eV, is attributed to the carbon atom linking with an oxygen atom of carbonyl group and and a single bonded oxygen atom (O–C=O) which exists in the oxidized hemicellulose and in the aldehyde group of oxidized glucose and the acetyl group of hemicellulose. C5, whose electron bonding energy is 283.5 eV, is attributed to the carbon silicon single bond (C-Si) which exists in the coupling agent treated BF but has negligible impurities in UBF. Since the intensity of combination of C4 and C5 is lower than that of C1, C2, and C3, the peak differentiating was primarily carried on C1, C2, and C3.

The oxygen atom on BF surface has three combination modes in terms of O1, O2, and O3. O1, whose electron bonding energy is 532–532.5 eV, is attributed to the O=C double bond. O2, whose electron bonding energy is 533–533.5 eV, is attributed to the O-C single bond. O3, whose electron bonding energy is 532–533 eV, is attributed to the O-Si single bond. The O3 peaks have much lower intensity than the other combined oxygen atoms and overlap with that of O1. The peak differentiating is primarily carried on O1 and O2.

The silicon atom on BF surface treated with silane coupling agent has two combination modes: Si1 and Si2. Si1, whose electron bonding energy is 100.5–101 eV, is attributed to the Si–C single bond. Si2, whose electron bonding energy is 102.5–103 eV, is attributed to the Si–O single bond.

Table 3 lists the XPS parameters obtained from the peak differentiated curves. It is indicated by Table 3 that the content of C1 increases and those of C2 and C3 decrease after the alkali treatment because the alkali treatment dramatically reduces the content of hemicellulose. It is indicated by the O1s differentiation results that the combination of C-O single bond attributed to the carbohydrate on the alkali treated BF decreases and the combination of C=O double bond increases. The ratio of O/C is almost unchanged but the content of O2 decreases, suggesting that most hemicellulose is removed by the alkali treatment.

The XPS parameters of the coupling agent treated BF is similar to that of UBF except that the Si content on the surface of ABF and ASBF are 2.38% and 4.15%, respectively. It is suggested that BF was fibrillated and had an enlarged effective interaction area after the alkali treatment. Therefore, the alkali treated BF can interact with more silane coupling agents covering the BF surface.

### 3.5. Thermogravimetric Analysis of BF

Figure 5 is the TGA curves and derivative thermogravimetric analysis (DTG) of BF modified by various methods. The TGA and DTG curves of UBF and SBF demonstrate three stages of weight loss, similarly to for typical natural fibers [49,50]. The first stage of weight loss is the loss of free water and crystal water [51]. The loss process continues until reaching 150 °C. The lost weight was approximately 6%. The second stage of weight loss is the thermal degradation of hemicellulose with low molecule weight and parts of amorphous cellulose. The temperature range of weight loss occurred between 150 °C and 330 °C. At this stage, the glucoside bonds of hemicellulose and cellulose inside BF broke where parts of C–O bonds and C–C bonds broke into low molecule weight volatiles and inflammable gas. Meanwhile, the lignin began to degrade. The third stage is the major stage of thermal degradation. The temperature range is between 330 °C and 600 °C. At this stage, the lignin and crystalline cellulose pyrolyzed and carbonized. The maximum weight loss rate is at 355 °C where the pyrolyzed lignin and crystalline cellulose produced a great deal of low molecule weight substances containing carbonyl and hydroxyl. The low molecule weight substances continued to react and dehydrate, which promoted the carbonization of cellulose.

The TGA curves of ABF and ASBF show two stages of thermal degradation: stage of water loss and the stage of degradation at high temperature. The difference between the alkali treated BF and the untreated BF is that the thermal degrading peak of hemicellulose and amorphous cellulose in ABF and ASBF disappeared, suggesting that the alkali treatment easily removed the impurities and the hemicellulose and amorphous cellulose in ABF and ASBF, which were degraded in temperatures ranging from 150 °C to 330 °C [52]. The thermal degradation data of BF modified by various methods are listed in Table 4. The onset degradation temperature (*T*_onset_) and the temperature at maximum weight loss (*T*_p_) of ABF are the lowest while those of ASBF are the highest in the compared bagasse fibers. *T*_onset_ and *T*_p_ of ASBF and SBF were higher than that of the other BF. This is attributed to the fact that the alkali treatment accelerated the degradation of ABF and ASBF by destroying their condensed structure, while the silane coupling agents enhanced the thermal stability of BF.

### 3.6. XRD of BF

Figure 6 is the XRD patterns of BF modified by various methods. The diffraction peaks at 2θ = 16° and 2θ = 22° are attributed to the crystalline cellulose [46]. The locations of these diffraction peaks are similar to each other, suggesting that the modification did not change the crystalline form. But the intensity of these diffraction peaks had diversified. For ABF, the diffraction peaks at 2θ = 16° and 2θ = 22° intensified and became sharp compared to that of UBF, indicating the increasing of crystallinity of cellulose. The reason for this is that alkali removed the amorphous cellulose and had few effects on the crystalline cellulose, which increased the content of crystalline cellulose. For SBF, the diffraction peaks at 2θ = 16° and 2θ = 22° weakened compared to that of UBF, indicating the decreasing of crystallinity of cellulose. For ASBF, the intensity of diffraction peaks at 2θ = 16° and 2θ = 22° is between those of ABF and SBF and approach that of UBF. This means that the effect of alkali treatment is competing with silanes modifications on the crystallinity of bagasse. It is suggested that the strength of natural fiber is associated with its condensed structure, including the crystalline morphology [33,53]. A higher crystallinity or more complete condensed structure of natural fiber such as BF facilitates the reinforcement of BF to the PLA matrix. How the microstructure of modified BF affects the mechanical properties of PLA/BF composites will be illustrated below.

### 3.7. POM of PLA/BF Composites

Figure 7 is the crystallization process of PLA/BF composites at 140 °C. The morphology of PLA/BF composites was recorded at 1 min, 5 min, 10 min, 20 min, and 30 min. As shown in Figure 7, the crystal nuclei appeared slowly in the pure PLA but appeared quickly in PLA/BF composites, implying that BF is an effective heterogeneous nucleating agent. So, the crystal nuclei formed easily around BF. Note that PLA/ASBF composites crystallized rather slowly in the compared composites, suggesting that the effective interfacial interactions inside PLA/ASBF composites seriously impeded the movement of PLA chains and therefore decreased the crystallizing rate of PLA/ASBF composites.

### 3.8. DSC of PLA/BF Composites

Figure 8 is the DSC curves of PLA/BF composites modified by various methods and the DSC data are listed in Table 5. For PLA/UBF composites and PLA/ABF composites, BF is an effective heterogeneous nucleating agent which promotes the nucleation rate but decreases the temperature of cold crystallization. For the PLA/SBF composites and PLA/ASBF composites, the modification of silane coupling agents has little effect on the glass transition temperature (*T*_g_) and melting temperature (*T*_m_), suggesting that the coupling agents had little effect on the intramolecular interactions and lamellar crystal structure. The silane coupling agents provided effective interfacial interactions between PLA and BF and restricted the movements of PLA chains. Therefore, the crystal growing rates of PLA/SBF composites and PLA/ASBF composites were slower than that of PLA. Meanwhile, the incorporated BF destroyed the regularity of PLA that made the crystallization of PLA/SBF composites and PLA/ASBF composites more difficult, resulting in the decreasing of crystallinity of the two composites. Moreover, PLA in PLA/ASBF composites had slower movement and lower crystallinity by comparison with that in PLA/SBF, suggesting that the alkali treatment caused the fibrillation of BF and the increasing of surface area of BF. Therefore, more BF in PLA/ASBF composites interacted with the silane coupling agents to provide stronger interfacial interactions than in PLA/SBF composites.

### 3.9. TGA of PLA/BF Composites

Figure 9 is the TGA and DTG curves of PLA/BF composites modified by various methods. The thermal degradation data are listed in Table 6. It can be seen from Table 6 that *T*_onset_ and *T*_p_ of pure PLA is 345.2 °C and 363.7 °C, respectively. *T*_onset_ and *T*_p_ of PLA/BF composites are all lower than that of pure PLA, suggesting that the incorporation of BF decreased the thermal stability of PLA. Compared with that of PLA/UBF, *T*_onset_ and *T*_p_ of PLA/SBF and PLA/ASBF show little variations while *T*_onset_ and *T*_p_ of PLA/ABF shows an obvious decrease, suggesting that the destroying of BF microstructure by the alkali treatment decreased the thermal stability of BF [45,49,50].

### 3.10. XPS of PLA/BF Composites

The PLA/BF composites primarily contain the elements in terms of carbon, hydrogen, and oxygen. After BF was modified by silane coupling agents, PLA/BF composites also contain silicon and nitrogen elements which are imported by the silane coupling agent. With the aim to explore how the coupling agents interact with bagasse, XPS was also used to investigate the chemical composition and combined valence state of elements in the interface of PLA/UBF composites, PLA/ABF composites, PLA/SBF composites, and PLA/ASBF composites. Figure 10 is the peak-differentiated characteristic curves of XPS for PLA, PLA/UBF, PLA/ABF, PLA/SBF, and PLA/ASBF composites. The corresponding parameters of XPS in terms of C, O, Si, and N elements are listed in Table 7.

It is indicated by Table 7 that the contents of C3 and O2 in PLA are much higher than those in the PLA-based composites, because PLA contains a large amount of carbon oxygen double bonds and carbon oxygen single bonds which derive from the ester groups of PLA. However in PLA/ABF and PLA/ASBF composites, the contents of C1 increase, the contents of C2 and C3 decrease, and the ratios of O/C are almost unchanged.

The electron binding energy of O1 of BF is about 532 eV which is attributed to the O=C double bond. As indicated by Table 7, the electron binding energy of O1 shifts to a higher value after the treatment of BF, suggesting that the carbonyl groups of PLA was affected by BF. It is confirmed that the hydrogen bonds between the carbonyl group and hydroxyl group make the electron binding energy of O1 shift to a higher value [54]. It is obvious that PLA also interacted with BF through hydrogen bonds which contribute to the improvement of interfacial compatibility.

The silicon atom has two bonding modes in terms of Si1 and Si2, as listed in Table 7. Si1 is attributed to Si–C single bond whose electron binding energy is 100.5–101 eV. Si2 is attributed to Si-O single bond whose electron binding energy is 102.5–103 eV.

In addition, the silane coupling agents have amino groups in SBF and ASBF filled PLA based composites. The nitrogen atom has one bonding mode whose electron binding energy is 399.5–400 eV, is attributed to the N-O single bond derived from the interaction of amino groups with carboxyl groups [55], as listed in Table 7.

On the surfaces of PLA/ABF and PLA/ASBF composites, the contents of Si are 1.78% and 2.83%, and those of N are 1.75% and 2.98%. It is suggested by the above XPS results that the content of Si after the sequence of the first alkali treatment and then the coupling agent treatment is higher than that when treated only by the silane coupling agent, which means the coupling after the former sequence is stronger than the coupling after the latter sequence. It is also indicated by the XPS results that there are N-O single bond on the surface of PLA/ASBF and PLA/SBF composites, as shown in Table 7, suggesting that the coupling agent combined on SF has also chemically bonded the carboxyl group of PLA. Moreover, the nitrogen content of PLA/ASBF composites is higher than that of PLA/SBF, implying that more N-O single bonds can form between ASBF and PLA.

Evidently, there are N-O single bonds and hydrogen bonds occurring on PLA/ASBF interface and on the PLA/SBF interface. In addition, there are also mechanical interlocking occurring on the PLA/ASBF interface and on the PLA/SBF interface. The formed N–O single bonds, hydrogen bonds, and mechanical interlocking provided PLA/ASBF and PLA/SBF with strong interfacial interactions and therefore improved the compatibility of PLA/ASBF and PLA/SBF composites. This is consistent with the report of Islam et al., in which the alkali treatment, amino-silane coupling agent modification, and mechanical interlocking on the interfaces provided PLA/hemp fiber composites with strong interfacial interactions [5]. In this work, the sequence of combination of alkali treatment with coupling agent treatment exhibited better improvement in the interfacial compatibility than the sequence of treatment by coupling agent.

### 3.11. Mechanical Properties of PLA/BF Composites

Table 8 shows the tensile properties of PLA/BF composites modified by various methods. The tensile strength of pure PLA is up to 73.5 MPa, which is 72.1% higher than that of PLA/UBF composites. But the tensile strength of PLA/ABF composites is 11.6% lower than that of PLA/UBF composites. The reason for this is that the alkali treatment destroyed the microstructure of BF in terms of decreasing the content of cellulose and decreasing the strength of cellulose besides the increasable interfacial compatibility between PLA and ABF [45]. The tensile strength of PLA/SBF composites and PLA/ASBF composites are obviously 33.1% and 15.2% higher than that of PLA/UBF composites. The reason for this is that the silane coupling agents decreased the hydrophilicity of SBF and ASBF and enabled the amino group to interact with the carboxyl group and hydroxyl group of PLAs [29]. The interfacial compatibility of PLA/SBF composites and of PLA/ASBF composites were enhanced and it was significant for the stress to transfer from PLA matrix to BF. As a result, the tensile strength of PLA/SBF and PLA/ASBF composites are higher than that of PLA/UBF and PLA/ABF.

Since the interfacial compatibility between BF and PLA gradually were enhanced, the tensile modulus PLA/BF composites gradually increases with upgraded modification. This means the tensile modulus of PLA/ASBF composites was greater than that of the PLA/SBF composites, which in turn was greater than that of PLA/ABF composites, which was greater than that of PLA/UBF composites. Due to the destructed microstructure of ABF, the tensile strength and modulus of ASBF/PLA is lower than that of PLA/SBF composites.

The variation of flexural properties of PLA/BF composites is similar to that of tensile properties, implying that the alkali treatment of BF decreased the strength of BF but the modification by silane coupling agents improved the interfacial compatibility between BF and PLA [33].

The impact strength of PLA/SBF composites and PLA/ASBF composites are 3.58 kJ/m^2^ and 3.29 kJ/m^2^, which are 25.0% and 15.0% higher than that of PLA/UBF composites, suggesting that the silane coupling agents dramatically improved the impact strength. The silane coupling agents are small molecule surfactants which can decrease the surface polarity of fillers and improve the dispersion of fillers. When the composites are impacted, the well interacted interfaces consume most impact energy, resulting in the increasing of impact strength.

It is demonstrated by the XPS results that the content of Si of PLA/ASBF is higher than that of PLA/SBF, implying that the alkali treatment had synergetic effect with coupling agent modification on the surface chemical properties. It has been demonstrated by Yah et al. [56] that the alkali treatment and coupling agent modification synergistically improved the mechanical properties of polypropylene/rice husk composites. In this work, the mechanical properties of PLA/ASBF composites is not distinctly higher than those of PLA/SBF composites, suggesting that the alkali treatment had no prominently synergetic effect with coupling agent modification on the mechanical properties of PLA/BF composites. The reason is that the destroyed microstructure of BF caused the decreasing of mechanical properties of PLA/BF composites. It is important to control the conditions for alkali treatment so that the strength of BF is well maintained. This issue will be further researched in future works.

### 3.12. SEM Morphology of PLA/BF Composites

Figure 11 is the SEM morphology of PLA/BF composites with different modified BF. There are cracks on the fractured surface of the pristine PLA, as shown in Figure 11a, indicating that the pristine PLA is brittle. Figure 11b is the fractured surface of PLA/UBF composites. UBF is pulled out of the PLA matrix, leaving caves on the fractured surface. The pulled out UBF is not stained by the PLA matrix. It is suggested that the interfacial compatibility between PLA and UBF is weak and PLA/UBF composites were destroyed by interfacial debonding. Similar destruction was observed in PLA/ABF composites. ABF is pulled out from the PLA matrix, leaving caves on the fractured surface and the pulled-out ABF was hardly stained by the PLA matrix, as seen in Figure 11c. To enhance the interfacial compatibility, silane coupling agent was used to modify BF. Figure 11d shows the fractured surface of PLA/SBF composites. SBF is embedded into PLA matrix and few pulled out SBF were observed. PLA/SBF was destroyed not by interface debonding but by fiber breaking and matrix cracking. It is suggested that silane coupling agent significantly enhanced the interfacial compatibility between SBF and PLA, as is also indicated in PLA/ASBF composites. Figure 11e shows good interfacial compatibility between ASBF and PLA. ASBF is better embedded into PLA matrix than SBF. No distinct interfacial gap is observed between ASBF and PLA. The fibrillated ASBF is stained with PLA. Fiber breaking and matrix cracking are observed on the fractured surface of PLA/ASBF composites. In a word, silane coupling agent played a more important role than alkali treatment did in improving the interfacial compatibility of PLA/BF composites. The combination of silane coupling agent treatment with alkali treatment synergistically improved the interfacial compatibility and therefore the mechanical properties of PLA/SBF composites.

## 4. Conclusions

The microstructure of BF was successfully tailored by alkali treatment and surface modification. The surface polarity of BF could be tailored by alkali treatment and surface modification. The alkali treatment removed the small molecule compounds, hemicellulose and parts of amorphous cellulose, resulting in the decrease of thermal stability and an increase of the crystallinity of BF. The surface energies of UBF, ABF, SBF, and ASBF were 19.8 mJ/m^2^, 34.7 mJ/m^2^, 12.3 mJ/m^2^, and 21.6 mJ/m^2^.

The surface of UBF was coarse and impure and the fiber strand inside UBF was tight. After alkali treatment, fiber fragments were spotted on ABF surface, indicating that ABF was severely fibrillated. The surface of SBF was rather smooth, but the molecular layer of coupling agents was not easily spotted.

Since the alkali treatment made ASBF fibrillated, the effective contact interfacial area of ASBF with the coupling agents increased and ASBF could interact with more coupling agents. As a result, the content of silicon element on the surface of ASBF was higher than that on the surface of ABF. The O/C ratios of UBF, ABF, SBF, and ASBF were 0.48, 0.53, 0.47, and 0.51, respectively. The content of silicon elements on the surface of ASBF (4.15%) was higher than that of SBF (2.38%) due to the synergistic effect of alkali treatment and silane coupling agent modification on the surface chemical properties.

As for the PLA/UBF composites and PLA/ABF composites, BF was an effective heterogeneous nucleating agent which promoted the crystallization rate and decreased the temperature of cold crystallization. The coupling agents had little effect on the intramolecular interactions and lamellar crystal structure. But the interfacial compatibility of PLA/SBF composites and PLA/ASBF were effectively improved by silane coupling agents, resulting in the decreasing of mobility of PLA chains which hindrance the crystallization of PLA/SBF composites and PLA/ASBF composites. Meanwhile, SBF and ASBF destroyed the regularity of PLA chains and thus led to the decreasing of crystallinity of PLA/SBF composites and PLA/ASBF composites. Moreover, PLA chains in PLA/ASBF composites had lower crystallinity by comparison with that in PLA/SBF. The silane coupling agent played a more important role than alkali treatment did in improving the interfacial compatibility of PLA/BF composites. However, the alkali treatment had no prominently synergetic effect with coupling agent modification on the mechanical properties of PLA/BF composites due to the destruction of alkali treatment on the microstructure of BF.

## Figures and Tables

**Figure 1 polymers-11-01567-f001:**
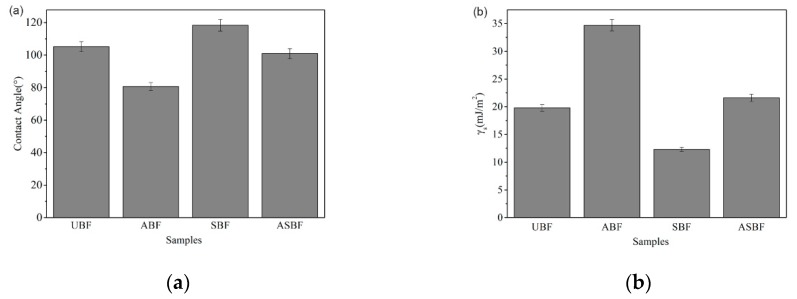
Surface polarity of bagasse fiber (BF) treated by different modification methods: (**a**) contact angle (**b**) surface energy.

**Figure 2 polymers-11-01567-f002:**
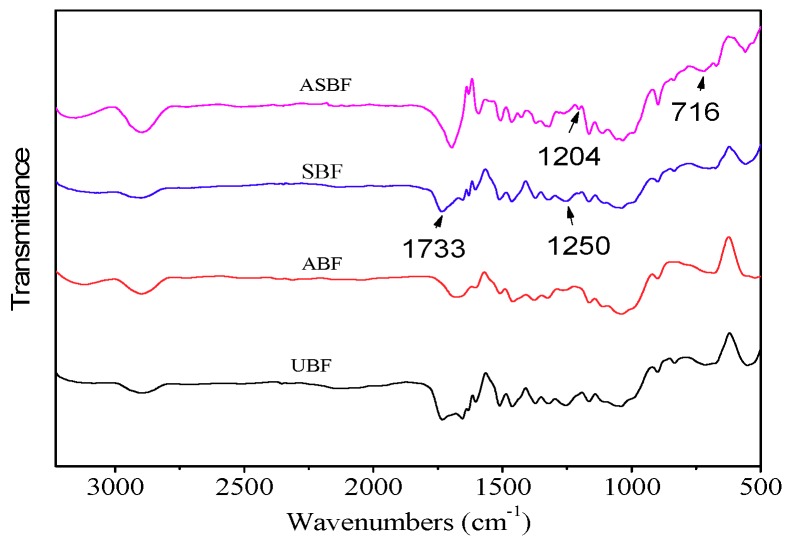
Fourier transform infrared spectroscopy (FTIR) spectra of BF modified by various methods.

**Figure 3 polymers-11-01567-f003:**
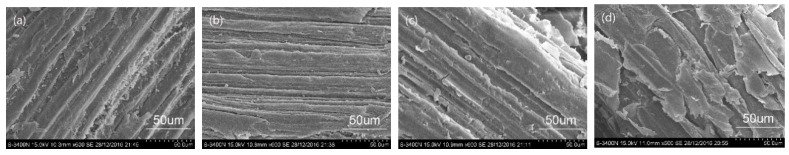
Scanning electron microscopy (SEM)morphology of BF modified by various methods: (**a**) untreated BF (UBF); (**b**) alkali treated BF (ABF); (**c**) BF modified by silane coupling agent (SBF); (**d**) BF modified combining alkali treatment with silane coupling agent (ASBF).

**Figure 4 polymers-11-01567-f004:**
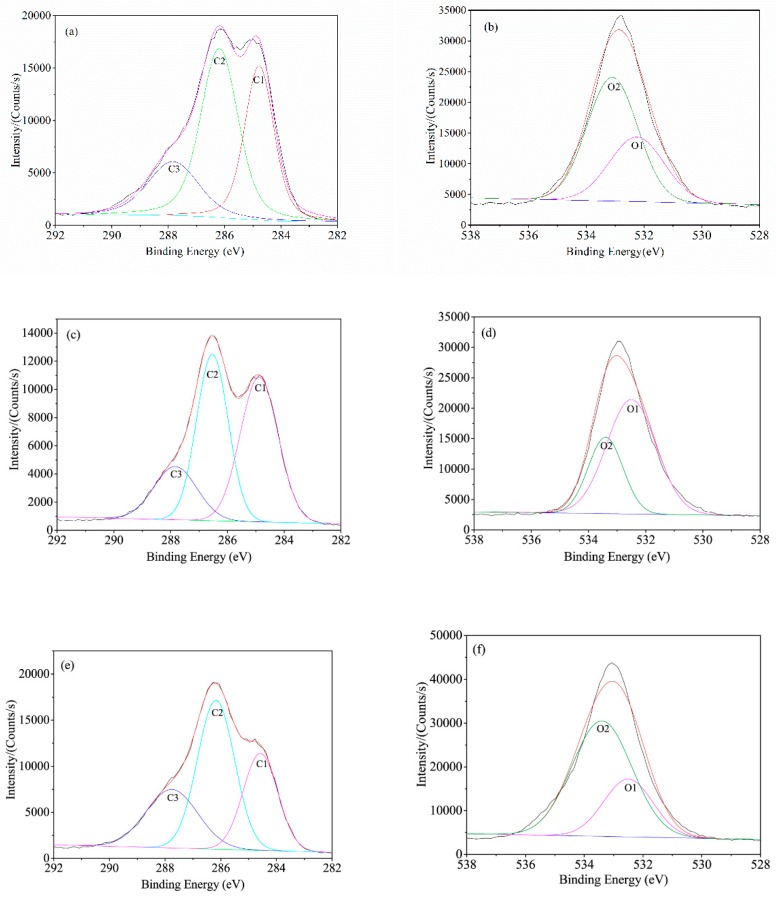
Peak differentiating curves of BF: (**a**) UBF-C1s; (**b**) UBF-O1s; (**c**) ABF-C1s; (**d**) ABF-O1s; (**e**) SBF-C1s; (**f**) SBF-O1s; (**g**) SBF-Si2p; (**h**) ASBF-C1s; (**i**) ASBF-O1s; (**j**) ASBF-Si2p.

**Figure 5 polymers-11-01567-f005:**
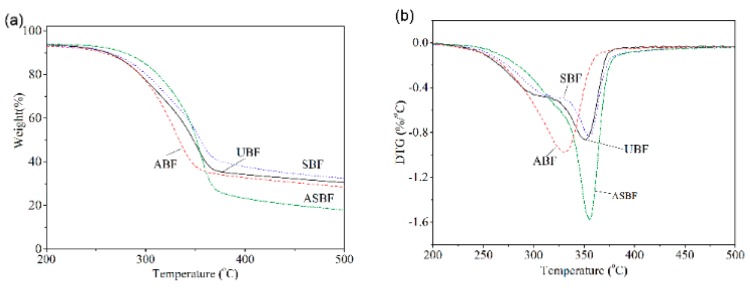
Thermogravimetry (TGA)curves of BF modified by various methods: (**a**) TGA, (**b**) DTG.

**Figure 6 polymers-11-01567-f006:**
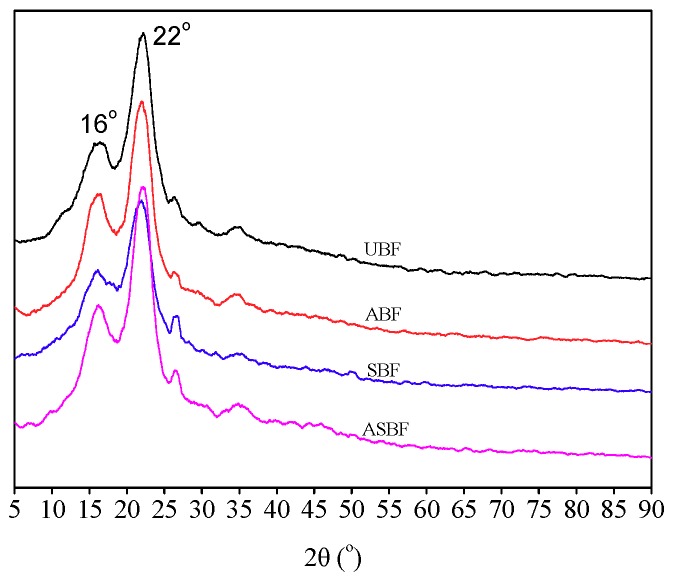
XRD patterns of BF modified by various methods.

**Figure 7 polymers-11-01567-f007:**
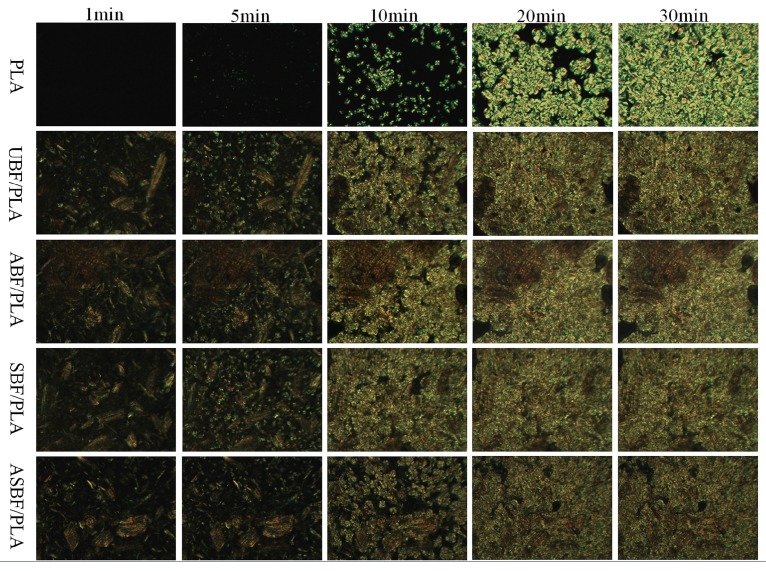
POM images of BF/PLA composites with different modified BF.

**Figure 8 polymers-11-01567-f008:**
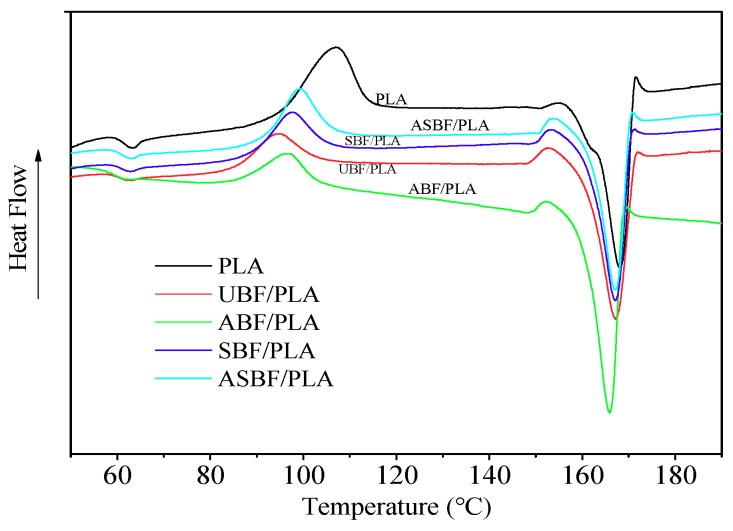
DSC curves of PLA/BF composites with different modified BF.

**Figure 9 polymers-11-01567-f009:**
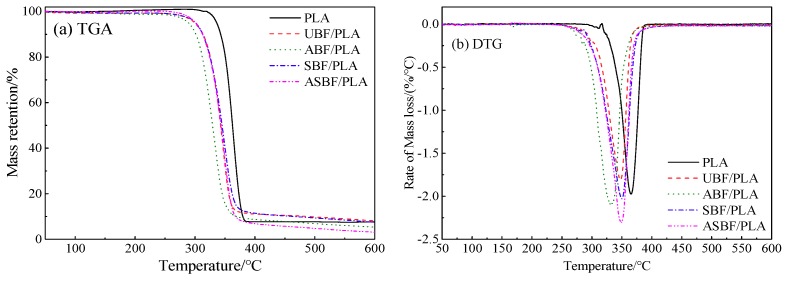
TGA curves of PLA/BF composites with different modified BF (Heating rate 10 °C/min).

**Figure 10 polymers-11-01567-f010:**
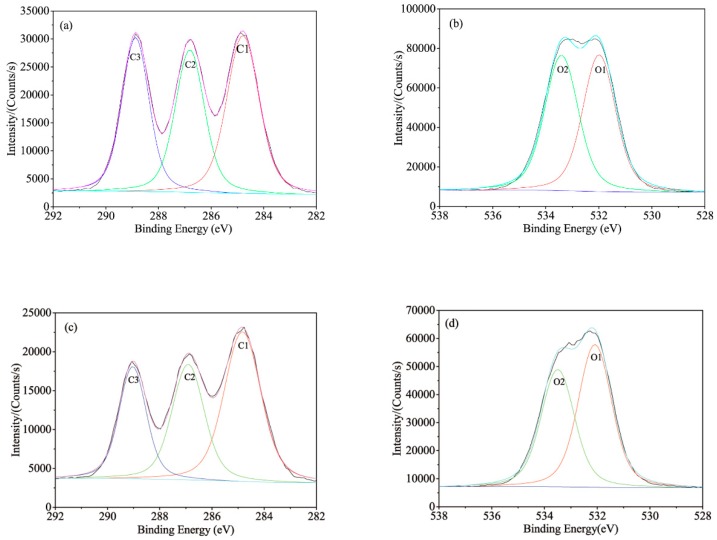
Peak differentiating curves of PLA/BF composites: (**a**) PLA-C1s; (**b**) PLA-O1s; (**c**) PLA/UBF-C1s; (**d**) PLA/UBF-O1s; (**e**) PLA/ABF-C1s; (**f**) PLA/ABF-O1s; (**g**) PLA/SBF-C1s; (**h**) PLA/SBF-O1s; (**i**) PLA/SBF-Si2p; (**j**) PLA/SBF-N1s; (**k**) PLA/ASBF-C1s; (**l**) PLA/ASBF-O1s; (**m**) PLA/ASBF-Si2p; (**n**) PLA/ASBF-N1s.

**Figure 11 polymers-11-01567-f011:**
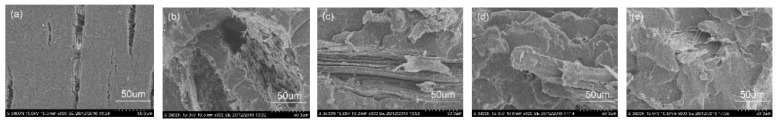
SEM images of fracture surfaces of PLA/BF composites: (**a**) PLA, (**b**) PLA/UBF, (**c**) PLA/ABF, (**d**) PLA/SBF, (**e**) PLA/ASBF.

**Table 1 polymers-11-01567-t001:** Components of PLA/BF composites.

Samples	Content (g)		BF Treatment
	PLA	BF	
PLA	100		
PLA/UBF	100	30	None
PLA/ABF	100	30	Alkali
PLA/SBF	100	30	Silane
PLA/ASBF	100	30	Alkali+silane

**Table 2 polymers-11-01567-t002:** Elements distribution on various BF surfaces.

Samples	C (Atomic%)	O (Atomic%)	Si (Atomic%)
UBF	49.665	49.695	0.640
ABF	48.121	51.341	0.538
SBF	51.698	46.268	2.034
ASBF	54.442	40.145	5.413

**Table 3 polymers-11-01567-t003:** XPS data of BF modified by various methods

Samples	Peak Location (eV)							C1s Peak Area (%)			O1s Peak Area (%)		O/C Ratio	Si Ratio (%)
	C1	C2	C3	O1	O2	Si1	Si2	C1	C2	C3	O1	O2		
UBF	284.70	286.08	287.60	532.25	533.10			31.26	43.85	24.89	36.07	63.93	0.48	
ABF	284.88	286.53	287.84	532.50	533.40			42.08	40.37	17.55	68.32	31.68	0.53	
SBF	284.58	286.17	287.76	532.50	533.40	101.50	102.90	28.20	46.35	24.45	33.04	66.96	0.47	2.38
ASBF	284.66	286.26	287.52	532.25	533.10	101.48	103.10	39.08	42.35	18.57	74.15	25.85	0.51	4.15

**Table 4 polymers-11-01567-t004:** Thermal degradation temperature of BF modified by various methods.

Samples	*T*_onset_/°C	*T*_p_/°C
UBF	291.4	350.7
ABF	282.1	330.4
SBF	300.6	354.7
ASBF	306.6	355.5

Note: *T*_onset_—Onset degradation temperature obtained from the intersection between the baseline and the tangent at maximum weight loss; *T*_p_—Temperature at maximum weight loss.

**Table 5 polymers-11-01567-t005:** Thermal properties of PLA/BF composites with different modified BF.

Samples	T_g_/°C	T_c_^1^/°C	∆H_c_^1^ (J/g)	T_m_/°C	∆H_m_/(J/g)	χ_c_/%
PLA	60.75	106.85	6.54	168.18	49.49	46.18
PLA/UBF	59.14	94.07	13.25	167.19	52.52	42.22
PLA/ABF	57.81	96.86	12.25	165.86	53.32	43.09
PLA/SBF	60.14	100.24	17.45	167.16	47.37	32.17
PLA/ASBF	61.59	99.02	19.39	167.18	45.63	26.24

Note: *T*_g_—The glass transition temperature. *T*_c_^1^—The temperature of cold crystallization. ∆H_c_^1^—The enthalpy of cold crystallization. *T*_m_—The melting temperature. ∆H_m_—The melting enthalpy.

**Table 6 polymers-11-01567-t006:** Mass loss temperatures of BF/PLA composites with different modified BF.

Samples	Heating Rate/(°C·min^−1^)	*T*_onset_/°C	*T*_p_/°C
PLA	10	345.2	363.7
PLA/UBF	10	320.6	347.8
PLA/ABF	10	306.8	331.4
PLA/SBF	10	322.0	349.6

Note: *T*_onset_—Onset temperature obtained from the intersection between the baseline and the tangent at maximum mass loss; *T*_p_—Temperature at maximum mass loss.

**Table 7 polymers-11-01567-t007:** XPS data of PLA/BF composites modified by various methods.

Samples	Peak Location (eV)								C1s Peak Area (%)			O1s Peak Area (%)		O/C Ratio	Si Ratio (%)	N Ratio (%)
	C1	C2	C3	O1	O2	Si1	Si2	N	C1	C2	C3	O1	O2			
PLA	284.81	286.79	288.84	532.03	533.34				37.73	34.60	27.67	50.71	49.29	0.74		
PLA/UBF	284.84	286.88	289.02	532.11	533.39				39.54	38.21	22.24	55.22	44.78	0.75		
PLA/ABF	284.87	286.83	288.98	532.14	533.39				45.64	33.39	20.97	54.29	45.71	0.73		
PLA/SBF	284.86	286.90	289.02	532.16	533.45	101.68	102.33	399.69	36.04	39.23	24.73	54.38	45.62	0.72	1.78	1.75
PLA/ASBF	284.84	286.75	288.87	532.06	533.30	101.64	102.21	400.06	44.27	34.25	21.48	56.83	43.17	0.75	2.83	2.98

**Table 8 polymers-11-01567-t008:** Mechanical properties of PLA/BF composites modified by various methods.

Samples	Impact Strength (kJ/m^2^)	Flexural Modulus (GPa)	Flexural Strength (MPa)	Tensile Modulus (GPa)	Tensile Strength (MPa)
PLA	2.88 ± 0.23	4.07 ± 0.16	112 ± 21	1.79 ± 0.21	73.5 ± 2.9
PLA/UBF	2.86 ± 0.25	5.06 ± 0.19	85 ± 16	2.03 ± 0.33	60.3 ± 5.8
PLA/ABF	2.67 ± 0.34	5.24 ± 0.21	78 ± 14	2.34 ± 0.29	53.2 ± 5.6
PLA/SBF	3.58 ± 0.27	5.24 ± 0.13	111 ± 19	2.57 ± 0.25	80.2 ± 6.7
PLA/ASBF	3.29 ± 0.36	5.16 ± 0.17	102 ± 22	2.62 ± 0.31	69.4 ± 7.5

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
