# Peer review of "Quantitively Characterizing the Chemical Composition of Tailored Bagasse Fiber and Its Effect on the Thermal and Mechanical Properties of Polylactic Acid-Based Composites"

_polymers, 2019, doi:10.3390/polym11101567_

Round 1

Reviewer 1 Report

The manuscript still needs some important changes/discussion prior to its final acceptance. Please find the following points below:

The meaning of some sentences are not clear, those sentences need to be restructured. Please check the introduction part and make it error-free. Herein, the final composites are the polylactide based bagasse fibers, so, the characterization like FTIR must be claimed after modification of BF using PLA and discuss what could be the changes that understand the fruitful preparation of the series of composites. Now, for the other characterizations, make it in a single figure and mark the changes after modifications. For example authors' used different captions for XPS and SEM analyses. Please make it regularly throughout the manuscript. The synergistic effect plays a vital role in the mechanical performances of the modified composites. So, a comparative study must be supplied to prove the justification. Fig. 11 is hard to follow, could you please elaborate the meaning? What does it mean by Ref. No. 5? 

Author Response

Dear Reviewer and Editor,

We have revised the manuscript according to your comments and suggestions. The revisions are indicated by green color.

The whole manuscript is seriously checked for spelling and grammars. We have tried to characterize the interfaces of PLA/BF composites by FTIR. But the resolution of FTIR is not high enough to offer any useful information on BF after modification by PLA. So we have to characterize and discuss the structure changes of BF by XPS. The changes after modification are well manifested not by the shifting of binding energy but by their Integral area of peaks. The integral area is not intuitive by comparing the curves in a single figure, so we have to compare the detailed changes after modifications characterized by XPS by tables listed as Table 3 and Table 7. Other changes on interfaces of PLA/BF composites are indicated on SEM figures. The comparative study is listed in the new Table (Table 8). Though the synergistic effect has been reported by Yeh et al, it is not manifest in the mechanical properties but in the structure characterization such as XPS in this report due to the destroying of microstructure of BF by alkali treatment. So we delete the descriptions that the alkali treatment has synergistic effect with silane coupling agent modification on the mechanical properties of PLA/BF composites. 11 is supplemented by Table 8 to indicate the changes of mechanical properties more clearly. The meaning of Ref. 5 is re-described.

We apricate for your important comments and suggestions. We also respect your opinions and seriously consider all the opinions you suggested. But we are not sure whether our revisions meet your requirement. If not, please let us know and give us opportunities to revise again.

Thank you for your comments and suggestions.

Sincerely Yours

Haoqun Hong

Reviewer 2 Report

This paper is Quantitively characterizing the chemical composition of tailored bagasse fiber and its effect on the thermal and mechanical properties of polylactic acid -based composites. The experiments are generally well performed and appear to be interesting. However, there are some questions as following.

1. Reference 5 is missing thing. Please provide the missing information for this reference.

2. Section 2 Additional details of the composite structures, specifically an indication of the bagasse fibers dimensions and their aspect ratio would be informative. Also, was there a preferred orientation of the reinforcing fibers in the composite structures? Reporting of these two factors would be useful.

3. The quantitative results of research can be added in abstract and conclusions section.

Author Response

Dear Reviewer and Editor,

We have revised the manuscript according to your comments and suggestions. The revisions are indicated by green color.

The meaning of Ref. 5 is re-described. Additional details of the composite structures are supplemented in Section 2. The quantitative results of research have been added in abstract and conclusions section.

Thank you for your comments and suggestions.

Sincerely Yours

Haoqun Hong

Round 2

Reviewer 1 Report

Now, the present form of the manuscript looks structured, scientifically elaborated, and has been accepted for the publication.